# Toxic Heavy Metals and Their Risk Assessment of Exposure in Selected Freshwater and Marine Fish in Thailand

**DOI:** 10.3390/foods12213967

**Published:** 2023-10-30

**Authors:** Prasit Sirisangarunroj, Nuntawat Monboonpitak, Weeraya Karnpanit, Piyanut Sridonpai, Alongkote Singhato, Nunnapus Laitip, Nattikarn Ornthai, Charun Yafa, Kunchit Judprasong

**Affiliations:** 1Master of Science Program in Nutrition, Faculty of Medicine Ramathibodi Hospital and Institute of Nutrition, Mahidol University, Nakhon Pathom 73170, Thailand; forgetnotnot@gmail.com; 2Institute of Nutrition, Mahidol University, Salaya, Phutthamonthon, Nakhon Pathom 73170, Thailand; nuntawat.mon@mahidol.ac.th (N.M.); piyanut.sri@mahidol.ac.th (P.S.); 3School of Science, Western Sydney University, Locked Bag 1797, Penrith, NSW 2751, Australia; w.karnpanit@westernsydney.edu.au; 4Nutrition and Dietetics Division, Faculty of Allied Health Sciences, Burapha University, Chonburi 20131, Thailand; alongkote@go.buu.ac.th; 5Chemical Metrology and Biometry Department, National Institute of Metrology (Thailand), Pathum Thani 12120, Thailand; nunnapusl@nimt.or.th (N.L.); nattikarn@nimt.or.th (N.O.); charun@nimt.or.th (C.Y.)

**Keywords:** toxic elements, risk assessment, fish, exposure

## Abstract

This study identified the levels of arsenic, cadmium, mercury, and lead in 15 species of commonly consumed fish in Thailand (7 freshwater species, 8 marine species), as well as the risk of these toxic elements for consumers. Inductively coupled plasma mass spectrometry (ICP-MS-MS) was used to identify toxic elements, while an exposure assessment was conducted by applying consumption amounts from the national food consumption survey to the toxic element contents in the fish samples. The results showed that the fish contained arsenic ranging from less than the limit of detection (LOD) to 8.51 mg/kg fresh weight (FW), cadmium ranging from the LOD to 0.04 mg/kg FW, and mercury ranging from the limit of quantitation (LOQ) to 0.38 mg/kg FW. Lead was found in small amounts (<LOQ) when compared to the Codex and Thailand’s standards. Only grouper had a higher mercury content (0.55 mg/kg FW) than that of the Codex standard, but it was lower than the notification requirement of Thailand’s Ministry of Public Health. Based on the estimated daily intake scenario, the consumption of most fish species posed a low risk of concern in terms of cadmium and lead. A high risk of concern was found for arsenic exposure, with the exception of long, non-scaly fish (catfish and dory). Marine fish, with the exception of Indo-Pacific mackerel, also posed a high risk of mercury exposure, but only in the case of a high mercury content, high consumption, or both. For a high arsenic content with high consumption or both, children aged 0–5.9 years were at a high risk of concern. Food safety authorities should regularly monitor the levels of toxic element contamination in high-risk food products.

## 1. Introduction

Fish have been consumed by the Thai people for at least 3000 years as evidenced by a striped head fish fossil [1]. As a food group, fish are important sources of high-quality proteins, minerals, vitamins, and essential omega-3 fatty acids [2], as well as being unique dietary sources for cardioprotection due to docosahexaenoic (DHA) and eicosapentaenoic (EPA) fatty acids [3]. Consequently, increased fish consumption is recommended to improve the health of all population groups.

Thailand’s economic growth has brought about extensive infrastructural and industrial development as well as the export industry promotion. Unfortunately, the absence of proper future planning has led to the deterioration of water sources and aquatic habitats due to the toxic elements used in industrial production (plastic, equipment, paint, PVC, and batteries), in agriculture (insecticide and fertilizer), and in medical/health industries (drugs, medical equipment, and cosmetics). While there are many types of toxic elements, arsenic (As), cadmium (Cd), lead (Pb), and mercury (Hg) are of priority concern in terms of food safety standards [4]. In particular, they play a prominent role in damaging the human nervous and urinary systems [4,5,6,7]. Long-term oral exposure to low levels of inorganic arsenic may cause dermal effects (such as hyperpigmentation and hyperkeratosis, corns, and warts) and peripheral neuropathy characterized by numbness in the hands and feet that may progress to a painful “pins and needles” sensation [5]. Cadmium in the human body can cause renal tubular damage, glomerular damage, decreased bone mineralization, increased risk of bone fractures, decreased lung function, and emphysema. These effects typically occur after long-term exposure to cadmium. Lead in the human body can affect decreased cognitive function, alterations in mood and behavior, altered neuromotor and neurosensory function, decreased glomerular filtration rate, increased blood pressure, decreased activity of several heme biosynthesis enzymes, decreased sperm, and spontaneous abortion [6]. Mercury is toxic to the nervous, digestive, and urinary systems. Therefore, Thailand’s Ministry of Public Health recognizes this danger and has mandated the maximum limit of As, Cd, Pb, and Hg in fish to be 2.0, 1.0, 0.3, and 0.5 mg/kg, respectively, based on the notification of Ministry of Public Health [8].

In response to consumer health concerns, research studies have investigated toxic-element contamination in commonly consumed freshwater and marine fish species in Thailand. For example, arsenic concentrations in Nile tilapia, red tilapia, striped snaked fish, king mackerel, sea bass, and grouper from commercial markets in Bangkok were reported to be 0.50, 1.76, 0.31, 3.11, 11.65, and 4.42 mg/kg, respectively, while mercury concentrations were <0.50, <0.50, 0.90, 1.13, 0.75, and 1.37 mg/kg, respectively [9]. Likewise, compared to the maximum limit, arsenic concentrations in king mackerel, sea bass, and grouper were reported to be higher than the maximum limit of 2 mg/kg, while mercury concentrations in striped snaked fish, king mackerel, sea bass, and grouper were higher than the maximum limit of 0.5 mg/kg [9].

Toxic element contamination in fish may be dependent on several factors, such as the location of aquatic zones, the contamination levels of fishing sites, environmental conditions, characteristics of the fish, and even household cooking practices. To explore this consumer health issue in more detail, this study investigated levels of arsenic, cadmium, mercury, and lead in commonly consumed fish using inductively coupled plasma mass spectrometry (ICP-MS-MS) and assessed the risk of exposure for Thailand’s consumers through the consumption of these fish species.

## 2. Materials and Methods

### 2.1. Sample Collection and Preparation

Fifteen commonly consumed fish species (seven freshwater and eight marine species) in Thailand were selected based on national food consumption survey data [10] and data in the food composition database [11]. Table 1 provides the common names, scientific names, and Thai names of the fish.

All species were purchased from three local fish markets within or near Bangkok (i.e., Bangkok-Noi market, Talad-Tai market, and Klong-Toey market). At each market, 3–4 vendors were randomly selected as representative of each source. Each fish was purchased, and the medium sizes were selected as representative of each species as presented in previous research [12]. All fish were weighed before and after removing inedible parts. All fish species with skin were prepared, homogenized, dried using a freeze-drier system, and analyzed for moisture content according to procedures published elsewhere [12,13]. After drying, the fish samples were blended into fine particles, kept in screw-capped plastic bottles, and stored at −20 °C until analysis.

### 2.2. Chemicals and Reagents

High-purity grade nitric acid (Suprapure 65% HNO_3_) was purchased from Sigma-Aldrich (St. Louis, MO, USA). Stock solutions of As, Cd, Hg, Pb, Rhodium (Rh), and certified reference material (CRM) SRM1566b (oyster tissue) were purchased from the National Institute of Standards and Technology (NIST), Gaithersburg, MD, USA. Another CRM, NMIJ7402-a (codfish tissue), was purchased from the National Metrology Institute of Japan (NMIJ). Milli-Q^®^ water (MilliporeSigma, Billerica, MA, USA) was used throughout the study.

### 2.3. Determination of Moisture

The moisture content of all samples was measured by drying the samples with sand in a water bath and then in a hot-air oven (Memmert ULE 400) at 100 ± 2 °C for 2 h, cooled in a desiccator, and weighed with a 4-digit analytical balance (Mettler AT201). The drying step was repeated for 1 h until a constant weight was reached (AOAC, 2019, method no. 925.23) [14].

### 2.4. Determination of Toxic Element Concentration

The concentrations of As, Cd, Hg, and Pb in samples were measured according to the AOAC Official Method 2015.01 (AOAC, 2019) [14]. Each sample, typically 0.25 g, was put into microwave digestion vessels. Concentrated HNO_3_, 4 mL, and 30% hydrogen peroxide (H_2_O_2_), 1 mL, were added into each vessel before the addition of internal standard Au + Lu solution 50 mg/L (Gold and Lutetium in 5% (*v*/*v*) HNO_3_) at 0.1 mL. Vessels were closed securely and placed in a microwave system. Each sample was digested at a minimum temperature of 190 °C for a minimum time of 10 min. After the digestion, the vessels were allowed to cool to room temperature and slowly opened with caution. Vessel contents were poured into an acid-cleaned 50 mL high-density polyethylene (HDPE) centrifuge tube and diluted with deionized water to a final volume of 20 mL. The toxic element contents in samples were measured using ICP-MS-MS (Agilent 8800 ICP-triple quadrupole MS, Agilent Technologies, Waldbronn, Germany). The concentrations of each toxic element in the fish samples were reported as mg/kg per fresh weight (FW). Toxic elements (As, Cd, Hg, and Pb) in the spiked samples were analyzed for method validation. The percentage recoveries of As, Cd, Hg, and Pb obtained were 90 ± 5% (85–95%), 100 ± 5% (95–105%), 111 ± 4% (107–115%), and 110 ± 4% (106–113%), respectively, which is in the acceptable range of 80–115%. The method limit of detection (LOD as 3SD of 10 times measurement of lowest concentration) of As, Cd, Hg, and Pb was 0.0007, 0009, 0.0011, and 0.0011 mg/kg FW, respectively. The method limit of quantitation (LOQ as 10SD of 10 times measurement of lowest concentration) of As, Cd, Hg, and Pb was 0.0063, 00062, 0.0124, and 0.0123 mg/kg FW, respectively.

### 2.5. Risk Assessment

Risk assessment was conducted by comparing the total exposure of individual toxic elements to their health-based guidance value (HBGV) as established by the Joint FAO/WHO Expert Committee on Food Additives (JECFA). Total exposure to each toxic element in the fish was calculated [15] using Equation (1):(1)$$TE=\frac{C\times DI\;}{BW}$$
where *TE* = total exposure of toxic element (μg/kg body weight), *C* = toxic element content in the sample (μg/g), *DI* = daily intake of fish (g/day), and *BW* = body weight (kg).

The risk as hazard quotient (HQ), expressed as a percentage of the HBGV, from exposure to individual toxic elements through fish consumption was calculated [15] using Equation (2). If the risk exceeded 100%, it indicated that the food was not safe for consumption.
(2)$$HQ=\frac{TE}{HBGV}\times P\times 100$$
where *HBGV* = health-based guidance value (μg/kg BW), which can be tolerable daily intake, tolerable weekly intake, or tolerable monthly intake depending on the toxic element; *P* = period (1 (daily), 7 (weekly), 30 (monthly)). *HBGV* of cadmium has a tolerable monthly intake of 25 μg/kg BW, and *HBGV* of mercury has a tolerable weekly intake of 25 μg/kg BW. On the other hand, *HBGVs* of lead and arsenic have been withdrawn [16]. Consequently, the above equation could not be used for arsenic; the margin of exposure (*MOE*) was used as an alternative indicator. *MOE* was calculated [15] using Equation (3). If the *MOE* was less than 100, it indicated that the consumption of the food might pose a risk or potential risk. On the other hand, if the *MOE* exceeded 100, it indicated a low risk of concern for public health.
(3)$$MOE=\frac{BMDL}{TE}$$
where *MOE* = margin of exposure, and *BMDL* = benchmark dose lower limit of inorganic arsenic (3 μg/kg bw per day) [16].

The risk from lead can be calculated using the interim reference limit with an adaptation of the equation. Total daily exposure to lead was calculated using Equation (4):(4)TDE=C×DI
where *TDE* = total daily exposure of toxic element (μg/day).

The risk from the consumption of fish was calculated using the following Equation (5). If the risk exceeded 100%, it indicated that the consumption of the food might pose a risk or potential risk.
(5)$$Risk=\frac{TDE}{IRL}\times 100$$
where *IRL* = interim reference limit (μg/day). The United States Food and Drug Administration (USFDA) has conducted a risk assessment of exposure to lead and proposed an interim reference limit (2.2 μg/day for children and 8.8 μg/day for women of childbearing age) [17].

### 2.6. Statistical Analysis

All measurements were performed in triplicate. The result of toxic element contents for each sample from the 3 markets was expressed as mean ± standard deviations (SD). One-way analysis of variance (ANOVA) and Duncan’s multiple range test were used to indicate significance of differences (*p* ≤ 0.05) among the mean values for each fish. Statistical analysis was performed using the SPSS version 18.0 Windows program (SPSS Inc., Chicago, IL, USA).

## 3. Results

### 3.1. Moisture and Toxic Element Contents

The moisture contents of the fifteen fish species ranged from 69.62 to 86.64% depending on the type of fish (Table 2). The total As, Cd, Hg, and Pb contents of the fish are shown in Table 2.

The arsenic content of freshwater fish ranged from 0.01 to 0.48 mg/kg fresh weight (FW). The three fish species containing the highest arsenic content were striped snake-head fish (0.48 ± 0.14 mg/kg FW), red tilapia (0.15 ± 0.10 mg/kg FW), and Nile tilapia (0.10 ± 0.07 mg/kg FW). For marine fish, the arsenic content ranged from 0.46 to 8.51 mg/kg FW, with the highest levels found in grouper (8.51 ± 1.60 mg/kg FW), long-tail tuna (1.92 ± 0.27 mg/kg FW), and king mackerel (1.36 ± 0.26 mg/kg FW).

The cadmium content in freshwater fish was lower than the limit of detection (<LOD, 0.0009 mg/kg FW), except for striped snake-head fish (<LOQ, 0.0062 mg/kg FW). For marine fish, the cadmium content was negligible in the range of LOD (0.0009 mg/kg FW) in seabass to 0.04 mg/kg FW in Indo-Pacific mackerel. The three highest cadmium contents were found in Indo-Pacific mackerel (0.04 ± 0.02 mg/kg FW), long-tail tuna (0.02 ± 0.02 mg/kg FW), and Atlantic mackerel (0.02 ± 0.01 mg/kg FW).

The mercury content in the freshwater fish was lower than the limit of quantitation (<LOQ, 0.0124 mg/kg FW), except for striped snake-head fish (0.02 ± 0.01 mg/kg FW). For marine fish, the mercury content was in the range of <LOQ in mullet to 0.38 mg/kg FW in grouper. The top three marine fish with highest mercury content were grouper (0.38 ± 0.17 mg/kg FW), king mackerel (0.09 ± 0.06 mg/kg FW), and long-tail tuna (0.07 ± 0.02 mg/kg FW).

All fish species contained lead in small amounts (<LOQ, 0.012 mg/kg FW).

### 3.2. Comparison of Toxic Element Content with Legal Standard

Most of the fish had toxic elements lower than the national standards [8], except for long-tail tuna and grouper that had total arsenic contents of 1.92 and 8.51 mg/kg FW, respectively, which are higher than the standard (2 mg/kg). Grouper also had a mercury content of 0.55 mg/kg FW, which is higher than the standard (0.5 mg/kg).

### 3.3. Risk Assessment of Toxic Elements through Fish Consumption

From the national food consumption data, the consumption amount was present in cooked form. In this study, the toxic element contents in the raw fish were low, and cooking could not affect these elements. Hence, the yield factor (weight before and after cooking) was used to calculate the toxic element content of the cooked samples. Different from the other toxic elements, one more important piece of information required for conducting a risk assessment for arsenic was the percentage of inorganic arsenic in fish. To calculate exposure for assessments, inorganic arsenic content must be used. The calculation of risk using the % inorganic arsenic of total arsenic (% iAs) as based on previous research [18,19,20] was applied. Fifteen types of fish were categorized into 5 food groups based on consumption data [21], with the yield factor and % inorganic arsenic of each group as shown in Table 3.

#### 3.3.1. Long Scaly Freshwater Fish

The results of this study revealed that persons aged 0–2.9 years old and those who had high consumption had a margin of exposure lower than 100, indicating a high risk of concern for arsenic (Table 4). Consumers in all age groups also had a margin of exposure lower than 100, which implies a high risk of concern for arsenic for consumers of long scaly freshwater fish. For cadmium, mercury, and lead, all population groups showed a percentage of risk less than 100, which implies safety from the risk of the consumption of long scaly freshwater fish.

#### 3.3.2. Long Non-Scaly Freshwater Fish

For arsenic, all population groups had a margin of exposure higher than 100, which implies a low risk of concern from arsenic for all populations. For cadmium, mercury, and lead, all population groups had a percentage of risk less than 100, which implies safety from the risk that can be caused by these elements from the consumption of long non-scaly freshwater fish.

#### 3.3.3. Flat Scaly Freshwater Fish

For arsenic (Table 5), the populations had a margin of exposure lower than 100, which implies a high risk of concern but only for some age groups (all age groups, except 18–64.9 years, for boiled fish and those aged 3–12.9 years for fried fish) and in extreme cases (97.5th percentile, high amount of consumption, and high arsenic content in fish). For consumers only, all age groups with consumption of high content of arsenic in fish, except for those persons aged 35 years or older for boiled fish and those aged 18 years or older for fried fish, had an MOE lower than 100, which implies a high risk of concern for arsenic. Some age groups (0–5.9 years old in boiled fish and 3–5.9 years old in fried fish) who had high consumption of average arsenic content in marine fish also had an MOE lower than 100, which can be assumed a high risk of concern from arsenic.

For cadmium (Cd), mercury (Hg), and lead (Pb), all groups of the population had a percentage of risk less than 100. This can be assumed as a low risk of concern that might be caused by cadmium, mercury, and lead from the consumption of flat scaly freshwater fish.

#### 3.3.4. Marine Fish

For arsenic (As) (Table 6), the results revealed that for the population per capita in all ages with a high amount of consumption, except the case of average arsenic amounts in marine fish at the age of 65 years and older in both boiled and fried fish and 13–17.9 years old in fried fish, the MOE was lower than 100, which can be assumed a high risk of concern from arsenic for eaters of marine fish. In addition, some age groups (0–5.9 years old in boiled fish and 0–2.9 years old in fried fish) with high consumption of an average mercury content in marine fish also had an MOE lower than 100, which can be assumed to be a high risk of concern from arsenic. Eaters in all age groups also had an MOE lower than 100, which can be assumed to be a high risk of concern from arsenic for eaters of marine fish.

For mercury (Hg) and population per capita, only the population aged 0–2.9 years old with high consumption of a high mercury content in marine fish had a percentage of risk lower than 100, which can be assumed as a low risk that might be caused by mercury from the consumption of marine fish. For eaters only, all age groups with consumption of a high content of mercury in fish, except those 65 years older in fried fish, had a percentage of risk lower than 100. Also, some age groups (0–12.9 years old in boiled fish and 0–2.9 years old in fried fish) with high consumption of an average mercury content in marine fish had a percentage of risk lower than 100. This can be assumed as a low risk that might be caused by mercury from the consumption of marine fish in cases of a high mercury content in marine fish.

For cadmium (Cd) and lead (Pb), all groups of the population had a percentage of risk less than 100. This can be assumed as safety from risk that can be caused by cadmium and lead from the consumption of marine fish.

#### 3.3.5. Indo-Pacific Mackerel

For arsenic (As), only the population per capita in the age of 3–5.9 years with a high amount of consumption and the age of 0–2.9 years in an extreme case (a high amount of consumption and a high arsenic content in fish) had an MOE lower than 100, which can be assumed a high risk of concern from arsenic. For eaters only, all ages with a high amount of consumption, except the age of 18–64.9 years with high consumption of an average mercury content in mackerel, had an MOE lower than 100 together with the age of 3–5.9 years with average consumption of a high arsenic content in mackerel, which can be assumed a high risk of concern from arsenic, as shown in Table 7.

For cadmium (Cd), mercury (Hg), and lead (Pb), all groups of the population had a percentage of risk less than 100. This can be assumed as safety from risk that can be caused by cadmium, mercury, and lead from the consumption of Indo-Pacific mackerel.

## 4. Discussion

### 4.1. Moisture and Toxic Element Contents

Most marine fish had arsenic and mercury content higher than freshwater fish, which agrees well with the study of Busamongkol et al. [9]. This may be due to the greater presence of toxic elements in the sea from ionic sources, causing the release of toxic elements in sediments [22]. The salinity of the ocean also affects the toxic element uptake of fish [23]. Of all marine fish, grouper had the highest arsenic and mercury content, which may be caused by their feeding habits (fish, crustaceans, and mollusks) [24]. This feeding habit can lead to biomagnification, which is a toxic element accumulated along the food chain [25]. The habitat of grouper, which is on the seafloor, unlike other fish except pomfret, may also play a part in the toxic element content [26,27,28,29,30,31,32] due to the toxic elements in sediments [33]. Pomfret’s feeding habit mostly consists of crustaceans [34], so it has lower biomagnification.

On the other hand, striped snakehead fish had the highest toxic element content of all freshwater fish. This may be due to their feeding habits and habitat. Striped snakehead fish is a carnivore whose food consists of fish, crustaceans, and insects [35], which can cause biomagnification [25], and it lives near the sediments of water [36], which can increase its toxic element intake. Toxic metal and metalloid contamination in swimming crab, shrimp, and squid from a eutrophic Brazilian estuary was reported, with crabs being the main bioaccumulators [37].

### 4.2. Comparison of Toxic Element in Fish with Legal Standard

As mentioned earlier, the total arsenic content in grouper and long-tail tuna was higher than the Thai standard value (2 mg/kg) [8], but this standard is specified only as inorganic arsenic, not total arsenic. Therefore, this study estimated the toxic element as inorganic arsenic (iAs), which was estimated as 10% of the total arsenic of fish [19], as shown in Table 3. The calculated results found that the estimated inorganic arsenic of long-tail tuna (0.19 mg/kg) and grouper (0.85 mg/kg) was lower than the Thai standard and may be considered a low risk of concern from arsenic. Grouper was also the only kind of fish that had its total mercury being slightly higher than the standard value (highest value at 0.55 mg/kg over the standard value at 0.5 mg/kg). Therefore, the consumption of grouper may cause hazards to human health from mercury content.

### 4.3. Risk Assessment of Toxic Elements through Fish Consumption

In the risk assessment of fish in this study (Table 4 and Table 5), long scaly freshwater fish had a high risk of concern from arsenic for eaters in all age groups. Long scaly freshwater fish is only a species of fish (striped snakehead fish), which means striped snakehead fish had a high risk of concern from arsenic for eaters only. Striped snakehead fish had no risk from cadmium, mercury, and lead, which agrees well with the study of Arampongpun [38], which also found that striped snakehead fish from the market in Bangkok was safe from cadmium’s adverse effect on human health.

Long non-scaly freshwater fish had a low risk of concern from arsenic and no risk from cadmium, mercury, and lead. The long non-scaly freshwater fish in this study were walking catfish and pangasius dory. Walking catfish and pangasius dory were safe for consumption, making them safe from the toxic element’s adverse effects. This result corresponds with the study of Juwa [39], which also found walking catfish from Kwan-Phayao to be safe from lead’s adverse effects on human health.

Flat scaly freshwater fish tended to have a high risk of concern from arsenic for eaters only in the case of a high arsenic content. The high arsenic content in this study at the 97.5th percentile of arsenic content in flat scaly freshwater fish was 0.25 mg/kg. This value was in the range of red tilapia (0.05–0.25 mg/kg) while the highest amount in other flat scaly freshwater fish was 0.17 mg/kg. Flat scaly freshwater fish had no risk from cadmium, mercury, and lead, similar to the study of Dokmaikaw and Suntaravitun [40], which also found that red tilapia from the Chachoengsao municipal market was safe from cadmium, mercury, and lead’s adverse effects on human health.

Marine fish had a high risk of concern from arsenic for all eaters only (Table 6 and Table 7) but no risk from cadmium and lead. In the case of mercury, marine fish had a high risk of concern from mercury in the case of a high mercury content in marine fish. The arsenic content at the 97.5th percentile in marine fish was 0.53 mg/kg, which was found in the range of grouper (0.22–0.55 mg/kg), while the highest amount in other marine fish was 0.15 mg/kg. It indicates that grouper may cause an adverse effect on consumer health from arsenic. This result is similar to the study of Thongra-ar [41], which found that marine fish from the coastal area of Map-ta-phut industrial estate also had risk from mercury’s adverse effects on human health but no risk from cadmium and lead.

Indo-Pacific mackerel had a high risk of concern from arsenic for eaters only when the eaters only had high consumption of Indo-Pacific mackerel, but there was no risk from cadmium, mercury, and lead. This result agrees with the study of Arbsuwan [42], which found that Indo-Pacific mackerel from the pier in the Khlong-yai district was safe from cadmium’s adverse effects on human health. Ritonga et al. [43] also found that Indian mackerel, belonging to the same family as Indo-Pacific mackerel from the market in Bangkok, was safe from mercury’s adverse effects on human health.

Most fish samples had a high risk of concern regarding arsenic exposure, except the long non-scaly freshwater fish group (Table 4 and Table 5). When specifying each age group in each fish group, ages 0–2.9 years and 3–5.9 years tended to have a lower margin of exposure than the other age groups. For example, when considering long scaly freshwater fish consumption, it posed a high risk of concern for ages 0–2.9 years, highlighting the need to assess the total risk from all fish groups. For certainty, the calculation of the margin of exposure was performed for each age group from all fish since consumers did not exclusively consume only one type of fish but rather consumed a variety of fish. The results showed that ages 0–2.9 years and 3–5.9 years had a margin of exposure of less than 100 (Table 4, Table 5, Table 6 and Table 7). This indicated that children aged 0–5.9 had a high risk of concern from arsenic exposure that requires special attention.

## 5. Conclusions

All studied toxic elements (As, Cd, Hg, and Pb) in the studied fish were found to be below the legal Thai standard, except grouper, which had the highest mercury content that did not comply with the legal standard. The risk assessment showed no risk from cadmium and lead in the studied fish. Most fish samples posed a high risk of concern regarding arsenic exposure, except the long, non-scaly freshwater fish group. The high-risk group of Indo-Pacific mackerel is mostly in an eaters-only group with high consumption while the high risk of flat-scaly freshwater fish only occurs when flat-scaly freshwater fish have a high arsenic content. On the other hand, the eaters-only group was identified as the high-risk group for marine fish, and long-scaly freshwater fish especially in children aged 0–5.9 years had a high risk of concern from arsenic. These results showed the possibility of the adverse effects of exposure to toxic elements from the consumption of the studied fish. Marine fish also had a high risk from mercury exposure for most of the eaters-only group, especially when consuming fish with a high mercury content. Information on potential foods contaminated with toxic elements should be provided to consumers to prevent adverse effects on human health. Food safety authorities should regularly monitor the levels of toxic element contamination in high-risk food products.

## Figures and Tables

**Table 1 foods-12-03967-t001:** The selected commonly consumed fish in this study.

English Name	Local Name	Scientific Name
Freshwater fish:		
Nile tilapia	Pla-Nil	*Oreochromis niloticus*
Striped snake-head fish	Pla-Chon	*Channa striatus*
Walking catfish	Pla-Duke	*Clarias batrachus*
Red tilapia	Pla-Tub-Tim	*Oreochromis niloticus-mossambicus*
Common silver barb	Pla-Ta-Pean	*Barbodes gonionotus*
Snakeskin gourami	Pla-Sa-Lid	*Trichogaster pectoralis*
Pangasius dory	Pla-Dol-Ly	*Pangasius hypophthalmus*
Marine fish:		
Indo-Pacific mackerel	Pla-Tu	*Rastrelliger brachysoma*
Mullet	Pla-Kra-Bok	*Mugil cephalus*
Sea bass	Pla-Ka-Pong-Khaw	*Lates calcarifer*
Longtail tuna	Pla-O	*Thunnus tonggo*
King mackerel	Pla-In-See	*Scomberomorus cavalla*
Grouper	Pla-Kao	*Epinephelus bruneus*
Silver pomfret	Pla-Ja-Ra-Med	*Pampus argenteus*
Atlantic mackerel	Pla-Sa-Ba	*Scomber scombrus*

**Table 2 foods-12-03967-t002:** Moisture contents and toxic element contents of each fish species; data expressed as mean ± SD ^1^ (*n* = 3).

Fish	Moisture (%)	Toxic Element Content (mg/kg Fresh Weight)			
		Arsenic	Cadmium	Mercury	Lead
Freshwater fish:					
Nile Tilapia (Pla Nil)	76.39 ± 0.67	0.10 ± 0.07 ^b^	ND ^2^	<LOQ ^3^	<LOQ ^3^
Pangasius dory (Pla Dolly)	86.64 ± 0.77	<LOQ ^3^	ND ^2^	<LOQ ^3^	<LOQ ^3^
Red Tilapia (Pla Tubtim)	73.64 ± 1.52	0.15 ± 0.10 ^b^	ND ^2^	<LOQ ^3^	<LOQ ^3^
Silver barb (Pla Tapean)	72.53 ± 3.39	0.02 ± 0.02 ^c^	ND ^2^	<LOQ ^3^	<LOQ ^3^
Snakeskin gourami (Pla Salid)	70.82 ± 3.36	0.02 ± 0.01 ^c^	ND ^2^	<LOQ ^3^	<LOQ ^3^
Striped snake-head fish (Pla Chon)	74.78 ± 0.63	0.48 ± 0.14 ^a^	<LOQ ^3^	0.02 ± 0.01	<LOQ ^3^
Walking catfish (Pla Duke)	69.62 ± 2.01	0.01 ± 0.00 ^c^	ND ^2^	<LOQ ^3^	<LOQ ^3^
Marine fish:					
Atlantic mackerel (Pla Saba)	75.56 ± 0.00	1.11 ± 0.24 ^cd^	0.02 ± 0.01	0.06 ± 0.04 ^b^	<LOQ ^3^
Grouper (Pla Kao)	78.76 ±1.42	8.51 ± 1.60 ^a^	0.01 ± 0.00	0.38 ± 0.17 ^a^	<LOQ ^3^
Indo-Pacific mackerel (Pla Tu)	75.57 ± 1.14	0.97 ± 0.16 ^cd^	0.04 ± 0.02	0.02 ± 0.01 ^b^	<LOQ ^3^
King mackerel (Pla Insee)	75.93 ± 0.33	1.36 ± 0.26 ^bc^	<LOQ ^3^	0.09 ± 0.06 ^b^	<LOQ ^3^
Long-tail tuna (Pla O)	71.34 ±2.22	1.92 ± 0.27 ^b^	0.02 ± 0.02	0.07 ± 0.02 ^b^	<LOQ ^3^
Mullet (Pla Krabok)	73.07 ± 5.45	0.46 ± 0.13 ^d^	<LOQ ^3^	<LOQ ^3^	<LOQ ^3^
Seabass (Pla Krapong)	74.87 ±0.38	1.32 ± 0.28 ^bc^	ND ^2^	0.02 ± 0.01 ^b^	<LOQ ^3^
Silver pomfret (Pla Jaramed)	76.60 ±0.10	0.61 ± 0.09 ^d^	0.01 ± 0.00	0.06 ± 0.02 ^b^	<LOQ ^3^

^1^ presented as mean ± SD from three individual markets (*n* = 3); ^2^ ND = not detected (less than LOD or 0.001 mg/kg); ^3^ <LOQ = less than LOQ (0.006 mg/kg for arsenic and cadmium or 0.012 for mercury and lead). Values with different superscript letters of species of fish in the same column were significantly different for a given variable (*p* < 0.05 one-way ANOVA followed by Duncan’s multiple range post hoc multiple comparisons).

**Table 3 foods-12-03967-t003:** List of fish in each food group, yield factor from cooking, and % inorganic arsenic (%iAS).

Group No.	Food Group	Fish	Yield Factor		%iAS
			Boiled	Fried	
Fresh water animals and products:					
1	Long scaly freshwater fish	- Striped Snake-head fish	0.93	0.68	23.6
2	Long non-scaly freshwater fish	- Walking Catfish - Pangasius dory	0.86	0.66	16.0
3	Flat scaly freshwater fish	- Nile Tilapia - Common Silver Barb - Red Tilapia - Snakeskin Gourami	0.86	0.63	13.0
Marine animals and products:					
1	Marine fish	- Seabass - King Mackerel - Mullet - Long-tail Tuna - Atlantic Mackerel - Grouper - Silver Pomfret	0.85	0.71	10
2	Indo-Pacific mackerel, purple-spotted bigeye and Hardtail scad	- Indo-Pacific mackerel	0.85	0.88	2.5

**Table 4 foods-12-03967-t004:** Margin of exposure of arsenic from long scaly freshwater fish consumption per capita.

Cooking Method	Type of Data		Margin of Exposure for Each Age Group						
	Food Consumption	Arsenic Content	0 to 2.9	3 to 5.9	6 to 12.9	13 to 17.9	18 to 34.9	35 to 64.9	65 and Older
Boiled	Average	Average	**57.7**	161.0	195.1	297.5	303.5	214.5	189.9
		97.5th percentile	**45.5**	126.8	153.7	234.4	239.0	169.0	149.6
	97.5th percentile	Average	**10.5**	**19.1**	**22.2**	**29.6**	**34.9**	**35.2**	**30.9**
		97.5th percentile	**8.3**	**15.0**	**17.5**	**23.3**	**27.5**	**27.7**	**24.3**
Fried	Average	Average	**79.3**	221.2	268.0	408.8	417.0	294.7	260.9
		97.5th percentile	**62.5**	174.2	211.1	322.0	328.4	232.2	205.5
	97.5th percentile	Average	**14.4**	**26.2**	**30.5**	**40.6**	**48.0**	**48.3**	**42.4**
		97.5th percentile	**11.4**	**20.7**	**24.0**	**32.0**	**37.8**	**38.1**	**33.4**

**Table 5 foods-12-03967-t005:** Margin of exposure of arsenic from flat scaly freshwater fish consumption for consumers only.

Cooking Method	Type of Data		Margin of Exposure of Arsenic in Each Age Group						
	Food Consumption	Arsenic Content	0 to 2.9	3 to 5.9	6 to 12.9	13 to 17.9	18 to 34.9	35 to 64.9	65 and Older
Boiled	Average	Average	171.9	128.7	188.3	226.7	283.2	338.8	369.7
		97.5th percentile	**51.9**	**38.8**	**56.8**	**68.4**	**85.5**	102.3	111.6
	97.5th percentile	Average	**87.6**	**60.2**	116.4	186.3	110.1	110.8	194.5
		97.5th percentile	**26.4**	**18.2**	**35.1**	**56.2**	**33.2**	**33.4**	**58.7**
Fried	Average	Average	236.6	177.1	259.2	312.1	389.9	466.4	509.0
		97.5th percentile	**71.4**	**53.5**	**78.2**	**94.2**	117.7	140.8	153.6
	97.5th percentile	Average	120.6	**82.8**	160.3	256.5	151.6	152.5	267.8
		97.5th percentile	**36.4**	**25.0**	**48.4**	**77.4**	**45.7**	**46.0**	**80.8**

**Table 6 foods-12-03967-t006:** Margin of exposure of arsenic from marine fish consumption per capita.

Cooking Method	Type of Data		Margin of Exposure of Arsenic for Each Age Group						
	Food Consumption	Arsenic Content	0 to 2.9	3 to 5.9	6 to 12.9	13 to 17.9	18 to 34.9	35 to 64.9	65 and Older
Boiled	Average	Average	234.7	417.2	705.1	1255.6	619.6	731.1	1057.6
		97.5th percentile	**50.1**	**89.0**	150.4	267.9	132.2	156.0	225.6
	97.5th percentile	Average	**17.6**	**40.3**	**77.9**	**86.6**	**43.9**	**74.2**	130.2
		97.5th percentile	**3.8**	**8.6**	**16.6**	**18.5**	**9.4**	**15.8**	**27.8**
Fried	Average	Average	279.3	496.4	839.0	1494.0	737.3	870.0	1258.4
		97.5th percentile	**59.6**	105.9	179.0	318.8	157.3	185.6	268.5
	97.5th percentile	Average	**20.9**	**47.9**	**92.7**	103.1	**52.2**	**88.3**	154.9
		97.5th percentile	**4.5**	**10.2**	**19.8**	**22.0**	**11.1**	**18.8**	**33.0**

**Table 7 foods-12-03967-t007:** Margin of exposure of arsenic from Indo-Pacific mackerel consumption for eaters only.

Cooking Method	Type of Data		Margin of Exposure of Arsenic for Each Age Group						
	Food Consumption	Toxic Element Content	0 to 2.9	3 to 5.9	6 to 12.9	13 to 17.9	18 to 34.9	35 to 64.9	65 and Older
Boiled	average	average	119.4	112.7	173.8	231.3	245.0	264.7	302.5
		97.5th percentile	104.3	**98.5**	151.8	202.0	214.0	231.2	264.2
	97.5th percentile	average	**66.4**	**29.7**	**57.5**	**92.1**	108.8	109.5	**96.1**
		97.5th percentile	**58.0**	**26.0**	**50.3**	**80.4**	**95.0**	**95.7**	**84.0**
Fried	average	average	116.4	109.9	169.4	225.4	238.8	258.1	294.8
		97.5th percentile	101.7	**96.0**	147.9	196.9	208.6	225.4	257.5
	97.5th percentile	average	**64.7**	**29.0**	**56.1**	**89.8**	106.1	106.8	**93.7**
		97.5th percentile	**56.6**	**25.3**	**49.0**	**78.4**	**92.6**	**93.2**	**81.9**

## Data Availability

Data are contained within the article.

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
