# Peer review of "Toxic Heavy Metals and Their Risk Assessment of Exposure in Selected Freshwater and Marine Fish in Thailand"

_foods, 2023, doi:10.3390/foods12213967_

Round 1
Reviewer 1 Report
Comments and Suggestions for Authors
-The introduction can be enriched with general information about each metal that highlights the importance of its study;
-Determination of Toxic Element Concentration:
Information regarding the percentage of recovery, limit of detection and quantification must be provided;
-The discussion about why the concentrations found can be enriched with justifications associated both with the habitat of these animals and with their migratory habits, differences in metabolism between species, among other parameters, as was briefly commented on in topic 4.1. Moisture and Toxic Element Contents. It should be noted that not all metals have the capacity to biomagnify throughout the entire trophic chain, some will only biomagnify between low trophic levels or may not have this capacity. These differences must be detected related to the species studied and the elements;
-It would be important to provide justifications for why younger age groups present a greater health risk;
-In a different approach to risk assessment, the Hazard quotient can be applied to verify in this modeling the possible health risk per element and in the end conduct the Hazard Index where the sum of each HQ of the elements studied can present the possibility of the existence of risk when considering the presence of all toxic elements present in the sample. The carcinogenic risk can also be applied to As. Below is an article where this approach was taken:
Toxic metal and metalloid contamination in seafood from an eutrophic Brazilian estuary and associated public health risks (https://doi.org/10.1016/j.marpolbul.2022.114367)
Author Response
Comments |
Response to the reviewers |
- The introduction can be enriched with general information about each metal that highlights the importance of its study; |
Thank you very much for your valuable comments. The information of each metal is added in the revised manuscript. |
- Determination of Toxic Element Concentration: Information regarding the percentage of recovery, limit of detection and quantification must be provided; |
The percentage of recovery, limit of detection and quantification are added in section “2.4. Determination of Toxic Element Concentration” of the revised manuscript. |
- The discussion about why the concentrations found can be enriched with justifications associated both with the habitat of these animals and with their migratory habits, differences in metabolism between species, among other parameters, as was briefly commented on in topic 4.1. Moisture and Toxic Element Contents. It should be noted that not all metals have the capacity to biomagnify throughout the entire trophic chain, some will only biomagnify between low trophic levels or may not have this capacity. These differences must be detected related to the species studied and the elements; |
Thank you very much for your valuable comments. Yes, certain elements may lack the capacity to biomagnify throughout the entire trophic chain; they might only show biomagnification within lower trophic levels or may not exhibit this capability at all. In our discussion, we have focused on providing a detailed explanation for arsenic and mercury. This is because these two elements were found at elevated concentrations in marine fish, particularly grouper, in comparison to their levels in freshwater fish. |
- It would be important to provide justifications for why younger age groups present a greater health risk; |
Because health-based guidance values such as PTWI for Hg, PTMI for Cd, and BMDL for iAs are calculated per kilogram of body weight, it is important to consider that children generally have lower body weights than adults. This difference in body weight increases the susceptibility of children to a greater risk from exposure to toxic metals. Additionally, it is worth noting that the IRL for Pb exposure, as established by the USFDA, is four times lower for children than for adults. As a result, a higher health risk in children has been observed. |
- In a different approach to risk assessment, the Hazard quotient can be applied to verify in this modeling the possible health risk per element and in the end conduct the Hazard Index where the sum of each HQ of the elements studied can present the possibility of the existence of risk when considering the presence of all toxic elements present in the sample. The carcinogenic risk can also be applied to As. Below is an article where this approach was taken: |
Yes, we used the hazard quotient to characterize the risk of exposure to Cd and Hg. We have revised Equation 2 by replacing the term 'Risk' with 'Hazard quotient (HQ).' HQ is commonly calculated for chemicals with health-based guidance values derived from the NOAEL/uncertainty factor. However, the health-based guidance values for As and Pb have been withdrawn. Therefore, we calculated the MOE equation using BMDL for As, and we used the updated IRL for Pb as a benchmark to evaluate whether As or Pb exposure from food is a potential concern. In line with the US EPA guidelines on cumulative risk assessment, the hazard index (HI) is a rough approximation for dose addition in the case of multiple chemical exposures. However, HI is typically applied to combine the hazards of compounds that share a common mechanism of toxicity, such as organophosphate pesticides and carbamates. Thus, to prevent an overestimation of exposure to multiple metals that do not belong to the same Common Mechanism Group (CMG), we did not perform a cumulative risk assessment. |
Toxic metal and metalloid contamination in seafood from an eutrophic Brazilian estuary and associated public health risks (https://doi.org/10.1016/j.marpolbul.2022.114367) |
Key information of this study is added in the revised manuscript. |

Reviewer 2 Report
Comments and Suggestions for Authors
The authors investigated the levels of arsenic, cadmium, mercury, and lead in commonly consumed fish using inductively coupled plasma mass spectrometry (ICP-MS-MS) and assessed the risk of exposure for Thailand’s consumers through consumption of these fish species. The results were amply discussed, and appear very promising. The reviewer found minimal issues in the work, whuch can be improved by the following:
a) In the introduction, specific to paragraph 1, include various fish types that are commonly/commercially sold in Thailand; paragraph 2, include why Thailand’s Ministry of Public Health takes this issue seriously; paragraph 3, include recommendations of the two studies cited in paragrph 3, use those recommendations to build your rationale of this current study in paragraph4.
b)Methods and results are very ok. In the discussion, please make sure to use "(Refer to Table ?)" in the places where data from the Tables2-7 are mentioned. That is ti say, all the Tables in results must be referred to at the discussion.
Look forward to your revised manuscript
Comments on the Quality of English Language
English language can be further improved
Author Response
Comments |
Response to the reviewers |
The authors investigated the levels of arsenic, cadmium, mercury, and lead in commonly consumed fish using inductively coupled plasma mass spectrometry (ICP-MS-MS) and assessed the risk of exposure for Thailand’s consumers through consumption of these fish species. The results were amply discussed, and appear very promising. The reviewer found minimal issues in the work, which can be improved by the following: |
|
a) In the introduction, specific to paragraph 1, include various fish types that are commonly/commercially sold in Thailand; paragraph 2, include why Thailand’s Ministry of Public Health takes this issue seriously; paragraph 3, include recommendations of the two studies cited in paragraph 3, use those recommendations to build your rationale of this current study in paragraph 4. |
Thank you very much for your valuable comments. Yes, I agreed with you. We did a sequence of the important of this study. |
b) Methods and results are very ok. In the discussion, please make sure to use "(Refer to Table ?)" in the places where data from the Tables 2-7 are mentioned. That is it say, all the Tables in results must be referred to at the discussion. |
All tables were added and indicated in the discussion section of the revised manuscript. |

Reviewer 3 Report
Comments and Suggestions for Authors
This manuscript aimed to evaluate the risk of arsenic, cadmium, mercury, and lead in fifteen species of commonly consumed fish in Thailand. However, I think the quality of this manuscript cannot be recommended in Foods. The comments are listed as follows:
1. The contents of heavy metals in fish were affected by various, including fish size, catching season, distribution area and individual differences. The measured results in this manuscript can not support the risk assessment of heavy metal uptake in fish in Thailand. First, the fish samples in this manuscript were collected in three local fish markets within or nearby Bangkok. Can such a limited area represent the fish in Thailand? Secondly, as the author told in Table 2, three samples were detected for one species of fish. The number is not enough to eliminate individual differences. Thirdly, the sizes of fish and catching season were not considered in this manuscript.
2. The title should revised to match the content of this manuscript.
Comments on the Quality of English Language
Moderate editing of English language required
Author Response
Comments |
Response to the reviewers |
This manuscript aimed to evaluate the risk of arsenic, cadmium, mercury, and lead in fifteen species of commonly consumed fish in Thailand. However, I think the quality of this manuscript cannot be recommended in Foods. The comments are listed as follows: |
Thank you very much for your valuable comments. However, we thought that this manuscript can fit to this journal in the Special Issue of “Heavy Metals and Potentially Toxic Elements (PTE) in Foods”.
|
1. The contents of heavy metals in fish were affected by various, including fish size, catching season, distribution area and individual differences. The measured results in this manuscript cannot support the risk assessment of heavy metal uptake in fish in Thailand. First, the fish samples in this manuscript were collected in three local fish markets within or nearby Bangkok. Can such a limited area represent the fish in Thailand? Secondly, as the author told in Table 2, three samples were detected for one species of fish. The number is not enough to eliminate individual differences. Thirdly, the sizes of fish and catching season were not considered in this manuscript. |
- I agreed with your comments. However, all samples in this study were collected from 3 wholesale markets which representative from all part of Thailand. Most of fish retailer in Thailand have to purchase in these wholesale markets. Therefore, this studied sample could represent the toxic level in Thai’s fish. - As results mentioned in section 3.1, toxic elements are presented as mean and standard deviation. When the concentration of toxic element in each specie above the LOQ level, it found in all 3 samples but different concentration (not three samples were detected for one species of fish). - The size of fish is a major factor for the toxic element concentration. More information is added in the revised manuscript as “Each fish was purchased and selected the medium sizes as representative of each species as presented in previous research [13]”. |
2. The title should revised to match the content of this manuscript. |
As your suggested, the title is modified as “Toxic Elements and their Risk Assessment of the Exposure in Selected Freshwater and Marine Fish”. |

Round 2
Reviewer 3 Report
Comments and Suggestions for Authors
The title should revised as: Toxic Heavy Metals and Their Risk Assessment of Exposure in Selected Freshwater and Marine Fish in Thailand
Comments on the Quality of English Language
Minor editing of English language required